# Meter Wave Polarization-Sensitive Array Radar for Height Measurement Based on MUSIC Algorithm

**DOI:** 10.3390/s22197298

**Published:** 2022-09-26

**Authors:** Guoxuan Wang, Guimei Zheng, Hongzhen Wang, Chen Chen

**Affiliations:** 1Graduate School, Air Force Engineering University, Xi’an 710051, China; 2Air Defense and Missile Defense College, Air Force Engineering University, Xi’an 710051, China

**Keywords:** meter wave radar, polarization-sensitive array, MUSIC, height measurement

## Abstract

Obtaining good measurement performance with meter wave radar has always been a difficult problem. Especially in low-elevation areas, the multipath effect seriously affects the measurement accuracy of meter wave radar. The generalized multiple signal classification (MUSIC) algorithm is a well-known measurement method that dose not require decorrelation processing. The polarization-sensitive array (PSA) has the advantage of polarization diversity, and the polarization smoothing generalized MUSIC algorithm demonstrates good angle estimation performance in low-elevation areas when based on a PSA. Nevertheless, its computational complexity is still high, and the estimation accuracy and discrimination success probability need to be further improved. In addition, it cannot estimate the polarization parameters. To solve these problems, a polarization synthesis steering vector MUSIC algorithm is proposed in this paper. First, the MUSIC algorithm is used to obtain the spatial spectrum of the meter wave PSA. Second, the received data are properly deformed and classified. The Rayleigh–Ritz method is used to decompose the angle to realize the decoupling of polarization and the direction of the arrival angle. Third, the geometric relationship and prior information of the direct wave and the reflected wave are used to continue dimension reduction processing to reduce the computational complexity of the algorithm. Finally, the geometric relationship is used to obtain the target height measurement results. Extensive simulation results illustrate the accuracy and superiority of the proposed algorithm.

## 1. Introduction

In recent years, with the emergence of stealth aircraft, anti-radiation missiles, and other weapons, meter wave radar, which can detect stealth targets and anti-radiation missiles, has received extensive attention [1,2]. However, due to meter wave radar’s wide beam, low band, and long wavelength, the ground-reflected echo cannot be ignored in low-elevation areas, which leads to the echo signal having a low signal-to-noise ratio and serious multipath coherence, thereby reducing the target detection ability of meter wave radar [3,4]. It is known that the essence of using radar to measure the target height is to estimate the target elevation angle first and to then calculate the target height according to the geometric relationship. Therefore, meter wave radar’s low-elevation estimation problem represents one of the key difficulties in the field of radar measurement.

During direction of arrival (DOA) estimation with low-angle targets, there is a serious multipath coherent signal in low-elevation areas. Additionally, there is a correlation and even coherence between the signal and the direct wave signal, which leads to the rank deficiency of the covariance matrix of the received data, destroys the orthogonality between the signal subspace and the noise subspace, and greatly reduces the accuracy of DOA estimation. Therefore, the conventional eigen-subspace super-resolution algorithm cannot accurately estimate the DOA of the source when there is a coherent source in space, and it requires decoherence preprocessing. The spatial smoothing algorithm is a commonly used decoherence pre-processing method [5]. The whole array is divided into multiple overlapping sub-arrays, and the covariance matrix of each sub-array is calculated and averaged to achieve decoherence. There are three smoothing methods: forward smoothing, backward smoothing, and forward-backward smoothing. However, the spatial smoothing algorithm has strong array requirements, and this algorithm loses the effective aperture of the array, resulting in a decrease in the accuracy of the algorithm. Additionally, the study in [6] shows that the spatial smoothing algorithm has almost no decoherence ability when the phase of the multipath attenuation coefficient is $0^{\circ }$ or $180^{\circ }$. The polarization smoothing algorithm [4] does not lose the array aperture because it makes full use of PSA polarization diversity technology. Moreover, by comparing polarization diversity technology with frequency diversity technology [3], it is found that when the phase difference is greater, the frequency diversity performance deteriorates sharply, while the polarization diversity does not have this disadvantage. Therefore, H. Kwak used the polarization smoothing technique to solve the influence of the multipath echo signal on the direct wave [7]. When the elevation angle was large, the accuracy was good, but when the elevation angle was small, the accuracy decreased sharply. This is because when the elevation is small, the difference in the reflection coefficient between horizontal polarization and vertical polarization is small. Then, Xu proposed a non-uniform weighting method for the autocorrelation matrix to improve the performance of the polarization smoothing algorithm [8]. However, the method only used the information from the autocorrelation matrix and did not achieve complete decoherence. Hence, the weighted polarization smoothing algorithm [9] was proposed to solve the problem, as this method can make full use of the autocorrelation and cross-correlation information of the sub-array output and can achieve better resolution performance and estimation accuracy. However, these methods have high computational complexity; thus, a method [10] combining the propagation operator with less calculation and polarization smoothing was proposed to solve the problem. In addition to polarization smoothing algorithms, various DOA estimation algorithms based on PSAs have also been proposed [11,12,13,14,15]. Multiple signal classification algorithms (MUSIC), the estimation of signal parameters via rotational invariance techniques (ESPRIT), maximum likelihood estimation (MLE), compressed-sensing approaches, and monopulse methods have been implemented in PSAs. However, these methods are based on uniform linear arrays (ULAs) and are only suitable for one-dimensional DOA estimation. For the uniform circular array (UCA), the tangential individually polarized UCA dipole algorithm [16] has been proposed, as it has a higher discrimination success probability, a higher estimation accuracy, and lower computational complexity than the ULA. In addition, the angle of the arrival estimation algorithm [17,18] based on the UCA also provides us with a reference. For DOA estimations of complex terrain, there is a broadband radar method based on a super-resolution algorithm [19]. Compared to conventional narrowband radar, this method has a better effect. The above decoherence algorithm can be applied to the meter wave radar height measurement of the polarization-sensitive array with appropriate deformation. For a direct height measurement model of a meter wave PSA, there are three main approaches: First, there is the polarization smoothing MUSIC algorithm [3] with the classical multipath signal model, which analyzes measurement performance. Second, there is a method that applies both the polarization smoothing algorithm and the spatial smoothing algorithm [20] for decoherence processing, solving the coherence between the direct wave and the reflected wave in low-elevation regions to a certain extent. Third, Tan proposed a method for elevation estimation using the generalized MUSIC algorithm after polarization smoothing [21]. It achieves better low-elevation angle estimation, but its computational complexity is still high, and the estimation accuracy and discrimination success probability need to be further improved. In addition, it cannot estimate the polarization parameters.

In order to further improve the height measurement accuracy and discrimination success probability of meter wave PSA radar in low-elevation areas, to reduce the computational complexity and to estimate the polarization parameters, the polarization synthesis steering vector MUSIC (P-SSV-MUSIC) algorithm is proposed in this paper. The proposed algorithm synthesizes the steering vector formed by the reflected wave to the direct waveguide vector and does not need to solve the coherence problem. Then, the MUSIC algorithm is used to obtain the elevation results of the target in a low-elevation area. The simulation results indicate that the proposed algorithm can achieve higher height measurement accuracy and discrimination success probability, can estimate the polarization parameters, and has lower computational complexity and a lower resolution threshold.

The rest of the paper is organized as follows: The multipath signal model of the meter wave PSA is given in Section 2. The polarization smoothing generalized MUSIC (PS-GMUSIC) algorithm is introduced in Section 3. The modified polarization smoothing generalized MUSIC (MPS-GMUSIC) algorithm is described in Section 4. The P-SSV-MUSIC algorithm is discussed in Section 5. The calculations of three the algorithms are given in Section 6. The correctness and effectiveness of the algorithm according to computer simulation results are proved in Section 7. Finally, Section 8 provides our conclusions.

## 2. Multipath Signal Model of Meter Wave PSA

Assume that a meter wave PSA radar system uses the classical multipath receiving signal model. The low-elevation reflection area is a smooth and flat surface, $f_{0}$ denotes the signal frequency, λ denotes the wavelength, and the PSA is composed of N biorthogonal dipoles that are arranged along the Z-axis and that are parallel to the X-axis and the Z-axis. As shown in Figure 1, $h_{a}$ and $h_{t}$ are the height of the PSA antenna and the height of the target, respectively; R is the horizontal distance between the point perpendicular to the ground and the radar antenna; $\theta_{d}$ is the incident angle of the target direct wave signal; $\theta_{s}$ is the incident angle of the target reflection multipath signal; and d is the element spacing.

Suppose that the target echo is a fully polarized electromagnetic wave and that φ is the target azimuth angle. We assumed that φ=π/2, which means that the target is incident in the YOZ plane; η∈[−π,π) is the polarization phase difference; and γ∈[0,π/2) is the polarization auxiliary angle. The polarization vector p in the Cartesian coordinate system of the meter wave PSA can be denoted as
(1)$$p(\theta ,\gamma ,\eta )=\left[\begin{array}{cc} \sin\theta \cos\varphi & -\sin\varphi \\ -\cos\theta & 0 \end{array}\right]\left[\begin{array}{c} \sin\gamma e^{j\eta }\\ \cos\gamma \end{array}\right]=\left[\begin{array}{c} -\cos\gamma \\ -\cos\theta \sin\gamma e^{j\eta } \end{array}\right]$$
where θ is the grazing angle of the direct signals and $\theta =\theta_{d}$ (Figure 1). The received signal for the PSA under multipath conditions can be expressed as
(2)$$x=\left(a(\theta_{d},\eta ,\gamma )+e^{-j\alpha }\rho a(\theta_{s},\eta ,\gamma )\right)s+n$$
where s is the echo signal vector after target scattering; n is the noise signal vector; $\alpha =4\pi h_{a}h_{t}/R\lambda$ is the phase difference generated by the delay difference between the reflected wave and the direct wave [22]; and $a(\theta_{d},\eta ,\gamma )$ and $a(\theta_{s},\eta ,\gamma )$ are the polarization spatial joint guidance vectors of the PSA corresponding to the direct wave and the ground-reflected wave, respectively, and are denoted as
(3)$$\{\begin{array}{c} \alpha \left(\theta_{d},\eta ,\gamma \right)=b\left(\theta_{d}\right)⊗p\left(\theta_{d},\eta ,\gamma \right)=\left[\begin{array}{c} -b\left(\theta_{d}\right)\cos\gamma \\ -b\left(\theta_{d}\right)\cos\theta_{d}\sin\gamma e^{j\eta } \end{array}\right]\\ \alpha \left(\theta_{s},\eta ,\gamma \right)=b\left(\theta_{s}\right)⊗p\left(\theta_{s},\eta ,\gamma \right)=\left[\begin{array}{c} -b\left(\theta_{s}\right)\cos\gamma \\ -b\left(\theta_{s}\right)\cos\theta_{s}\sin\gamma e^{j\eta } \end{array}\right] \end{array}$$
where ⊗ denotes the Kronecker product, and $b(\theta_{d})$ and $b(\theta_{s})$ are the steering vectors for the direct wave and the ground-reflected wave. These can be expressed as
(4)$$\{\begin{array}{c} b(\theta_{d})={\left[1,e^{-j2\pi d\sin\left(\theta_{d}\right)/\lambda },...e^{-j2\left(N-1\right)\pi d\sin\left(\theta_{d}\right)/\lambda }\right]}^{T}\\ b(\theta_{s})={\left[1,e^{-j2\pi d\sin\left(\theta_{s}\right)/\lambda },...e^{-j2\left(N-1\right)\pi d\sin\left(\theta_{s}\right)/\lambda }\right]}^{T} \end{array}$$

ρ is the Fresnel reflection coefficient’s vector matrix. which contains the horizontal and vertical polarization reflection coefficients and can be denoted as
(5)$$\rho =\mathrm{diag}(\underset{N}{\underset{︸}{\rho_{h}\cdots \rho_{h}}},\underset{N}{\underset{︸}{\rho_{v}\cdots \rho_{v}}})$$
where $\rho_{h}$ and $\rho_{v}$ are the Fresnel reflection coefficients for horizontal and vertical polarization, respectively. These can be defined as follows [23]:(6)$$\{\begin{array}{c} \rho_{h}=\frac{\sin\theta_{d}-\sqrt{\varepsilon -\cos^{2}\theta_{d}}}{\sin\theta_{d}+\sqrt{\varepsilon -\cos^{2}\theta_{d}}}\\ \rho_{v}=\frac{\varepsilon \sin\theta_{d}-\sqrt{\varepsilon -\cos^{2}\theta_{d}}}{\varepsilon \sin\theta_{d}+\sqrt{\varepsilon -\cos^{2}\theta_{d}}} \end{array}$$
where ε is the complex permittivity of the smooth ground. It can be expressed by $\varepsilon_{r}$ and $\sigma_{e}$ as follows [23]:(7)$$\varepsilon =\varepsilon_{r}-j60\lambda \sigma_{e}$$
where $\varepsilon_{r}$ represents the relative permittivity, and $\sigma_{e}$ is the conductivity of the reflective surface.

As we can see from Equation (6), the Fresnel reflection coefficients are determined by $\theta_{d}$ and ε. Hence, we can obtain $\rho_{h}\approx \rho_{v}\approx -1$ when the grazing angle θ is very low. In other words, there is very little difference between $\rho_{h}$ and $\rho_{v}$ when the grazing angle is very low. We use vector C to define the synthetic steering vector of the PSA for simplicity, with C being denoted as
(8)$$C=\left(a(\theta_{d},\eta ,\gamma )+\psi a(\theta_{s},\eta ,\gamma )\right)$$
where ψ is the fading coefficient vector of the received signal. It can be expressed as
(9)$$\psi =e^{-j\alpha }\rho$$

By substituting Equation (8) into Equation (2), the received signal in Equation (2) can be simplified as
(10)x=Cs+n

Additionally, the covariance of the received signal in Equation (10) is:(11)$$R=\sigma_{s}^{2}CC^{H}+I_{N}\sigma_{n}^{2}$$
where the superscript ${\left(\cdot \right)}^{H}$ denotes conjugate transpose; $I_{N}$ is an N×N identity unit matrix; and $\sigma_{s}^{2}=E\left[ss^{H}\right]$ and $\sigma_{n}^{2}=E\left[nn^{H}\right]$ are the covariance of the signal and noise, respectively. We defined the signal-to-noise ratio of the received signal as $\mathrm{SNR}=\sigma_{s}^{2}/\sigma_{n}^{2}$.

## 3. PS-GMUSIC Algorithm

The PS-GMUSIC algorithm combines polarization smoothing and the generalized MUSIC algorithm for PSA via the following procedure [21]:

Firstly, according to the horizontal polarization component signal and the vertical polarization component signal received by the PSA [3], the received signals for the PSA on the coordinate’s axis can be expressed as follows:(12)$$\{\begin{array}{c} x_{h}=x_{X}=A\left[\begin{array}{c} 1\\ \psi_{h} \end{array}\right]\left(-\cos\gamma \right)s+n_{h}=Aa_{h}\left(-\cos\gamma \right)s+n_{h}\\ x_{v}=x_{Z}=A\left[\begin{array}{c} 1\\ \psi_{v} \end{array}\right]\left(-\sin\gamma \right)\cos\theta_{d}e^{j\eta }s+n_{v}=Aa_{v}\left(-\sin\gamma \right)\cos\theta_{d}e^{j\eta }s+n_{v}\; \end{array}$$
where $n_{h}$ and $n_{v}\;$ are noise signals; $A=\left[b(\theta_{d}),b(\theta_{s})\right]$ is the composite steering vector that contains $b(\theta_{d})$ and $b(\theta_{s})$; $a_{h}={\left[1,\psi_{h}\right]}^{T}$ and $a_{v}={\left[1,\psi_{v}\right]}^{T}$; and $\psi_{h}$ and $\psi_{v}$ are the horizontal and vertical polarization fading coefficients, respectively. Additionally, $\psi_{h}$ and $\psi_{v}$ are expressed as
(13)$$\{\begin{array}{c} \psi_{h}=\rho_{h}e^{-j\alpha }\\ \psi_{v}=\rho_{v}e^{-j\alpha } \end{array}$$

According to Equation (11) and Equation (12), the covariance matrices of the multipath signals received via vertical polarization and horizontal polarization can be expressed as
(14)$$\{\begin{array}{c} R_{h}=E\left[x_{h}x_{h}^{H}\right]=\sigma_{s}^{2}\cos^{2}\gamma Aa_{h}a_{h}^{H}A^{H}+I_{N}\sigma_{n}^{2}=AR_{sh}A^{H}+I_{N}\sigma_{n}^{2}\\ R_{v}=E\left[x_{v}x_{v}^{H}\right]=\sigma_{s}^{2}\sin^{2}\gamma \cos^{2}\theta_{d}Aa_{v}a_{v}^{H}A^{H}+I_{N}\sigma_{n}^{2}=AR_{sv}A^{H}+I_{N}\sigma_{n}^{2} \end{array}$$

Second, the covariance matrix requires polarization smoothing. Polarization smoothing is an effective decoherence algorithm that can remove the correlation between the direct wave and the ground-reflected wave. The process of polarization smoothing [4] can be summarized as
(15)$$R^{ps}=\left(R_{h}+R_{v}\right)/2=\sigma_{s}^{2}AQ_{ps}A^{H}+I_{N}\sigma_{n}^{2}$$
where $Q_{ps}$ is the signal envelope matrix of the covariance matrix $R_{ps}$ after polarization smoothing. This can be derived as follows:(16)$$Q_{ps}=\frac{1}{2}\left(Q_{h}+Q_{v}\right)=\frac{\cos^{2}\gamma }{2}\left[\begin{array}{cc} 1 & \psi_{h}^{*}\\ \psi_{h} & {\left|\psi_{h}\right|}^{2} \end{array}\right]+\frac{\sin^{2}\gamma \cos^{2}\theta_{d}}{2}\left[\begin{array}{cc} 1 & \psi_{v}^{*}\\ \psi_{v} & {\left|\psi_{v}\right|}^{2} \end{array}\right]$$

In Equation (15), $R_{h}$ and $R_{v}$ can be estimated as follows
(17)$$\{\begin{array}{l} R_{h}=\frac{1}{T}\sum_{t=1}^{T}x_{h}(t)x_{h}{\left(t\right)}^{H}\\ R_{v}=\frac{1}{T}\sum_{t=1}^{T}x_{v}(t)x_{v}{\left(t\right)}^{H} \end{array}$$
where T is the number of received signal snapshots.

Third, we need singular value decomposition for covariance matrix B after polarization smoothing, which has the following form:(18)$$R^{ps}=U_{s}^{ps}\mathrm{diag}\left(\lambda_{1},\lambda_{2}\right){\left(U_{s}^{ps}\right)}^{H}+U_{n}^{ps}\mathrm{diag}\left(\lambda_{3},\cdots ,\lambda_{N}\right){\left(U_{n}^{ps}\right)}^{H}$$
where $\lambda_{1}>\lambda_{2}\geq \lambda_{3}≅\lambda_{4}\cdots \lambda_{N}$ is the eigenvalue; $U_{s}^{ps}$ is the eigenvector corresponding to two larger eigenvalues, representing the signal subspace; and $U_{n}^{ps}$ is the eigenvector group corresponding to (N−2) smaller eigenvalues, representing the noise subspace. Therefore, we can obtain the noise subspace projection matrix P, which is
(19)$$P=U_{n}^{ps}{\left(U_{n}^{ps}\right)}^{H}$$

Next, we can obtain the spatial spectrum of the PS-GMUSIC algorithm as follows:(20)$$P_{PS-\mathrm{GMUSIC}\;}\left(\theta_{d},\theta_{s}\right)=\frac{\det\left(A{\left(\theta_{d},\theta_{s}\right)}^{H}A\left(\theta_{d},\theta_{s}\right)\right)}{\det\left(A{\left(\theta_{d},\theta_{s}\right)}^{H}PA\left(\theta_{d},\theta_{s}\right)\right)},\theta_{d}>0^{{}^{\circ}},\theta_{s}<0^{{}^{\circ}}$$
where det[⋅] denotes the determinant of the matrix.

As we can see from Equation (20), it requires a two-dimensional angle search, which increases the computational complexity of the algorithm. In order to reduce the computational complexity, we can use the prior information from the multipath model (Figure 1): the geometric relationship [24] between the direct wave $\theta_{d}$ and the reflected wave $\theta_{s}$:(21)$$\theta_{s}=-\arctan\left(\tan(\theta_{d})+\frac{2h_{a}}{R}\right)\approx -\theta_{d}$$

Substituting Equation (21) into Equation (20), the two-dimensional angle search in Equation (20) can be simplified to a one-dimensional angle search:(22)$$P_{PS-\mathrm{GMUSIC}\;}\left(\theta_{d}\right)=\frac{\det\left(A{\left(\theta_{d},-\theta_{d}\right)}^{H}A\left(\theta_{d},-\theta_{d}\right)\right)}{\det\left(A{\left(\theta_{d},-\theta_{d}\right)}^{H}PA\left(\theta_{d},-\theta_{d}\right)\right)},\theta_{d}>0^{{}^{\circ}}$$

Finally, we need to convert the estimation of the low-elevation angle to the height of the target after conducting the spectral peak search. The conversion formula is as follows:(23)$$H\approx R_{r}\sin\theta_{d}+h_{a}$$
where $R_{r}$ denotes the distance between the target and the radar.

## 4. Proposed MPS-GMUSIC Algorithm

As we can see from Equation (16), the decoherence ability of polarization smoothing in low-elevation regions is related to the difference in the ground reflection coefficient between vertical polarization and parallel polarization. The rank of the signal envelope matrix $Q_{ps}$ after polarization smoothing is 2 when $\psi_{h}\neq \psi_{v}$, which indicates that the polarization smoothing algorithm can remove the coherence between the direct wave and the ground reflection wave. However, the rank of the signal envelope matrix $Q_{ps}$ after polarization smoothing is 1 when $\psi_{h}=\psi_{v}$. Additionally, the difference in the ground reflection coefficients between vertical polarization and parallel polarization is very small in low-elevation areas, and their values are all approximately −1. Therefore, the polarization smoothing decoherence ability is poor at this time, resulting in the meter wave PSA having poor angle measurement accuracy in low-elevation areas. In order to improve the decoherence performance of the polarization smoothing algorithm, we propose the MPS-GMUSIC algorithm, which is modified after polarization smoothing. The essence of the modified processing is forward and backward spatial smoothing processing with the number of subarrays being 1 [24]. The process of the MPS-GMUSIC algorithm is as follows:(24)$$R^{psm}=R^{ps}+I_{v}{\left(R^{ps}\right)}^{*}I_{v}=\sigma_{s}^{2}AQ_{ps}A^{H}+\sigma_{s}^{2}I_{v}A^{*}Q_{ps}^{*}A^{T}I_{v}+2I_{N}\sigma_{n}^{2}$$
where the superscript (⋅)* denotes complex conjugation, and $I_{v}$ denotes the inverse diagonal unit matrix, which is:(25)$$I_{v}={\left[\begin{array}{ccc} 0 & \cdots & 1\\ \vdots & ∴ & \vdots \\ 1 & \cdots & 0 \end{array}\right]}_{N\times N}$$

According to the definition of the composite steering vector A and $I_{v}$, the following relation equations can be obtained: (26)$$\{\begin{array}{l} I_{v}A^{*}=A\left[\begin{array}{cc} e^{j2\pi (N-1)d\sin\theta_{d}/\lambda /} & 0\\ 0 & e^{j2\pi (N-1)d\sin\theta_{s}/\lambda } \end{array}\right]\\ A^{T}I_{v}=\left[\begin{array}{cc} e^{-j2\pi (N-1)d\sin\theta_{d}/\lambda } & 0\\ 0 & e^{-j2\pi (N-1)d\sin\theta_{s}/\lambda } \end{array}\right]A^{H} \end{array}$$

By substituting Equation (26) into Equation (24), $R^{psm}$ in Equation (24) can be simplified as:(27)$$R^{psm}=\sigma_{s}^{2}AQ_{ps}A^{H}+\sigma_{s}^{2}AQ_{ps}^{*}A^{H}+2I_{N}\sigma_{n}^{2}=\sigma_{s}^{2}AQ_{psm}A^{H}+2I_{N}\sigma_{n}^{2}$$
where $Q_{psm}$ is the signal envelope matrix of the modified polarization smoothing covariance matrix. It is defined as follows:(28)$$Q_{psm}=Q_{ps}+Q_{ps}^{*}=\cos^{2}\gamma \left[\begin{array}{cc} 1 & {\bar{\psi }}_{h}\\ {\bar{\psi }}_{h} & {\left|\psi_{h}\right|}^{2} \end{array}\right]+\sin^{2}\gamma \cos^{2}\theta_{d}\left[\begin{array}{cc} 1 & {\bar{\psi }}_{v}\\ {\bar{\psi }}_{v} & {\left|\psi_{v}\right|}^{2} \end{array}\right]$$
where ${\bar{\psi }}_{h}$ is the real part of the horizontal polarization multipath attenuation coefficient $\psi_{h}$, and ${\bar{\psi }}_{v}$ is the real part of the vertical polarization multipath attenuation coefficient $\psi_{v}$. It can be seen from Equation (28) that when ${\bar{\psi }}_{h}\neq {\bar{\psi }}_{v}$, the rank of the signal envelope matrix $Q_{psm}$ after modified polarization smoothing is 2, which means that the modified polarization smoothing algorithm can also effectively remove the coherence between the direct wave and the ground reflection wave. Additionally, the modified polarization smoothing covariance matrix can be estimated as follows:(29)$$\hat{R^{psm}}=\hat{R^{ps}}+I_{v}{\left(\hat{R^{ps}}\right)}^{*}I_{v}$$
where $\hat{R^{psm}}$ is the polarization smoothing covariance matrix. Then, we need to conduct singular value decomposition for $\hat{R^{psm}}$ via the following form:(30)$$\hat{R^{psm}}=U_{s}^{psm}\mathrm{diag}\left(\lambda_{1},\lambda_{2}\right){\left(U_{s}^{psm}\right)}^{H}+U_{n}^{psm}\mathrm{diag}\left(\lambda_{3},\cdots ,\lambda_{N}\right){\left(U_{n}^{psm}\right)}^{H}$$
where $\lambda_{1}>\lambda_{2}\geq \lambda_{3}≅\lambda_{4}\cdots \lambda_{N}$ is the eigenvalue; $U_{s}^{psm}$ is the eigenvector corresponding to two larger eigenvalues, representing the signal subspace; and $U_{n}^{psm}$ is the eigenvector group corresponding to (N−2) smaller eigenvalues, representing the noise subspace. Therefore, we can obtain the noise subspace projection matrix $P_{psm}$, which is
(31)$$P_{psm}=U_{n}^{psm}{\left(U_{n}^{psm}\right)}^{H}$$

Next, we can obtain the spatial spectrum of the MPS-GMUSIC algorithm as follows:(32)$$P_{PS-\mathrm{MGMUSIC}\;}\left(\theta_{d},\theta_{s}\right)=\frac{\det\left(A{\left(\theta_{d},\theta_{s}\right)}^{H}A\left(\theta_{d},\theta_{s}\right)\right)}{\det\left(A{\left(\theta_{d},\theta_{s}\right)}^{H}P_{psm}A\left(\theta_{d},\theta_{s}\right)\right)},\theta_{d}>0^{{}^{\circ}},\theta_{s}<0^{{}^{\circ}}$$

Substituting Equation (21) into Equation (32), we can obtain:(33)$$P_{MPS-\mathrm{GMUSIC}\;}\left(\theta_{d},\theta_{s}\right)=\frac{\det\left(A{\left(\theta_{d},-\theta_{d}\right)}^{H}A\left(\theta_{d},-\theta_{d}\right)\right)}{\det\left(A{\left(\theta_{d},-\theta_{d}\right)}^{H}P_{psm}A\left(\theta_{d},-\theta_{d}\right)\right)},\theta_{d}>0^{{}^{\circ}},\theta_{s}<0^{{}^{\circ}}$$

As we can see from Equation (33), the two-dimensional spectral function becomes a one-dimensional spectral function, which greatly reduces the amount of calculation.

In the end, we need to use Equation (23) to convert the estimated low-elevation angle to the height of the target after the peak search. From the above two methods, it can be seen that the advantages of the PS-GMUSIC algorithm and the MPS-GMUSIC algorithm are that they are independent of the reflection coefficient and have strong robustness to the position. However, the disadvantages are that the accuracy is not high enough, and the polarization parameters cannot be estimated. Therefore, we propose the P-SSV-MUSIC algorithm, which has higher estimation accuracy and can estimate both the incident angle of the target direct wave signal $\theta_{d}$ and the polarization parameters γ,η. It also has low algorithm complexity.

## 5. Proposed P-SSV-MUSIC Algorithm

As we can see from Equation (8), the signal synthesis steering vector C is one-dimensional, and its rank is 1, expressing that the reflected wave polarization airspace joint steering vector is synthesized into the direct wave polarization airspace joint steering vector. Additionally, a rank of 1 means that the array-receiving signal model only has one incident signal source. Therefore, many conventional super-resolution DOA estimation algorithms can be directly applied to this signal model. It is necessary to demonstrate that the classical MUSIC algorithm has good directional resolution characteristics for incoherent signals as well as good measurement accuracy. In this paper, the conventional MUSIC algorithm is applied to the meter wave polarization-sensitive array elevation model, and dimension reduction is also carried out. The whole process can be derived as follows:

First, we need to decompose the eigenvalues of Equation (10) and to divide its eigenvector. Then, the only eigenvector corresponding to a large eigenvalue is used to constitute the signal subspace ${\bar{E}}_{s}$, and its 2M×(2M−1)-dimension eigenvector is used to constitute the noise subspace ${\bar{E}}_{n}$. After this, according to the conventional MUSIC algorithm, the spatial spectrum of the meter wave polarization-sensitive array can be obtained as follows:(34)$$P(\theta_{d},\theta_{s},\rho_{h},\rho_{v},\gamma ,\eta )=\frac{1}{{\left[C(\theta_{d},\theta_{s},\rho_{h},\rho_{v},\eta ,\gamma )\right]}^{H}{\bar{E}}_{n}{\bar{E}}_{n}{}^{H}C(\theta_{d},\theta_{s},\rho_{h},\rho_{v},\eta ,\gamma )}$$

As we can see from Equation (34), it contains six unknowns and requires six-dimensional search processing, which is not suitable for practical engineering applications. Therefore, we need to reduce the number of dimensions. We divided the dimensionality reduction into two stages. In the first stage, we decoupled the polarization information and the DOA information to reduce the dimensionality. In the second stage, we used the relationship between the direct wave and reflected wave and the relationship between the reflection coefficient and the direct wave to reduce the number of dimensions.

In the first stage of dimension reduction, the polarization information and DOA information need to be decoupled to reduce the dimensionality, so the signal model in Section 2 needs to be deformed and classified. First, we rewrite Equation (1) in the following form:(35)$$p(\theta ,\gamma ,\eta )=\left[\begin{array}{c} E_{X}\\ E_{Z} \end{array}\right]=\left[\begin{array}{cc} \sin\theta \cos\varphi & -\sin\varphi \\ -\cos\theta & 0 \end{array}\right]\left[\begin{array}{c} \sin\gamma e^{j\eta }\\ \cos\gamma \end{array}\right]=\left[\begin{array}{c} -\cos\gamma \\ -\cos\theta \sin\gamma e^{j\eta } \end{array}\right]=Ε(\theta )g(\gamma ,\eta )$$
where Ε(θ) and g(γ,η) are defined as follows:(36)$$Ε(\theta )\overset{\mathrm{def}}{=}\left[\begin{array}{cc} \sin\theta \cos\varphi & -\sin\varphi \\ -\cos\theta & 0 \end{array}\right],\varphi =\pi /2$$
(37)$$g(\gamma ,\eta )\overset{\mathrm{def}}{=}\left[\begin{array}{c} \sin\gamma e^{j\eta }\\ \cos\gamma \end{array}\right]$$

Therefore, Equation (8) can be transformed into
(38)$$C=b(\theta_{d})⊗\left[Ε(\theta_{d})g(\gamma ,\eta )\right]+e^{j\alpha }b(\theta_{s})⊗\left[Ε(\theta_{s})\rho g(\gamma ,\eta )\right]$$

Next, we classify the reflection coefficient and the wave path difference into one class, the steering vector into another class, and the polarization information into the same category. After this, Equation (38) can be further deformed to obtain
(39)$$C=\left[\begin{array}{cc} b(\theta_{d})⊗Ε(\theta_{d}) & b(\theta_{s})⊗Ε(\theta_{s}) \end{array}\right]\left[\begin{array}{c} I\\ e^{j\alpha }\rho \end{array}\right]g(\gamma ,\eta )$$

Since the reflection coefficient ρh,ρv and range difference α are functions of θd,θs, we can obtain the following definition:(40)D(θd,θs)=[b(θd)⊗Ε(θd)b(θs)⊗Ε(θs)]∈ℂ2M×4
(41)Df(ρh,ρv)=[Iejαρ]
(42)D(θd,θs,ρh,ρv)=D(θd,θs)Df(ρh,ρv)

According to Equations (40)–(42), Equation (39) can be simplified as follows:(43)$$\begin{array}{ll} C(\theta_{d},\theta_{s},\rho_{h},\rho_{v},\eta ,\gamma ) & =D(\theta_{d},\theta_{s},\rho_{h},\rho_{v})g(\gamma ,\eta )\\ & =D(\theta_{d},\theta_{s})D_{f}(\rho_{h},\rho_{v})g(\gamma ,\eta ) \end{array}$$

Then, according to Equation (43), we can define the MUSIC cost function as:(44)$$V={\left[D(\theta_{d},\theta_{s},\rho_{h},\rho_{v})g(\gamma ,\eta )\right]}^{H}{\bar{E}}_{n}{\bar{E}}_{n}{}^{H}D(\theta_{d},\theta_{s},\rho_{h},\rho_{v})g(\gamma ,\eta )$$

It is not difficult to find $g{\left(\gamma ,\eta \right)}^{H}g(\gamma ,\eta )=1$. Therefore, Equation (44) can be further transformed into
(45)$$V=\frac{{\left[D(\theta_{d},\theta_{s},\rho_{h},\rho_{v})g(\gamma ,\eta )\right]}^{H}{\bar{E}}_{n}{\bar{E}}_{n}{}^{H}D(\theta_{d},\theta_{s},\rho_{h},\rho_{v})g(\gamma ,\eta )}{g{\left(\gamma ,\eta \right)}^{H}g(\gamma ,\eta )}$$

Through the observation of Equation (45), it can be found that Equation (45) conforms to the criteria for the maximum and minimum Rayleigh quotients [25]. Therefore, we can obtain
(46)$$\lambda_{min}\left(\theta_{d},\theta_{s}\right)=\underset{g\left(\gamma ,\eta \right)\neq 0}{\min}\left(V\right)$$
where $\lambda_{\min}\left(\theta_{d},\theta_{s}\right)$ denotes the minimum eigenvalue obtained by the eigendecomposition of the matrix $D{\left(\theta_{d},\theta_{s},\rho_{h},\rho_{v}\right)}^{H}{\bar{E}}_{n}{\bar{E}}_{n}{}^{H}D(\theta_{d},\theta_{s},\rho_{h},\rho_{v})$. Therefore, we can obtain the four-dimensional search equation, which is
(47)$$f_{4D-\mathrm{MUSIC}}=\arg\underset{\theta_{d},\theta_{s},\rho_{h},\rho_{v}}{\max}[1/\lambda_{\min}(\theta_{d},\theta_{s},\rho_{h},\rho_{v})]$$

The eigenvector corresponding to the minimum eigenvalue is the estimated value of the polarization parameter $\hat{g}(\gamma ,\eta )$, so the polarization parameter can also be obtained by the following formula:(48)$$\left\{ \begin{array}{l} \hat{\gamma }=\arctan\frac{{\left[\hat{g}(\gamma ,\eta )\right]}_{1}}{{\left[\hat{g}(\gamma ,\eta )\right]}_{2}}\\ \hat{\eta }=∠\frac{{\left[\hat{g}(\gamma ,\eta )\right]}_{1}}{{\left[\hat{g}(\gamma ,\eta )\right]}_{2}} \end{array}\right.$$

As we can see from Equation (47), the polarization information and DOA information have been decoupled, which reduces the two search dimensions. However, the number of calculations required for a four-dimensional search element is still large. The second stage of dimension reduction is carried out below:

First, by substituting Equation (21) into Equation (47), Equation (47) can be reduced to a three-dimensional angle search.

Secondly, as we can see from Equation (6), the reflection coefficient $\rho_{h},\rho_{v}$ is determined by the incident angle $\theta_{d}$, the relative dielectric constant $\varepsilon_{r}$, and the surface material conductivity $\sigma_{e}$. Additionally, the relative dielectric constant $\varepsilon_{r}$ and the surface material conductivity $\sigma_{e}$ are different and are in different positions, such as in water, land, and vegetation. Fortunately, previous scholars have measured and summarized specific values under different scenarios [23], as shown in Table 1. According to Table 1, the relative dielectric constant $\varepsilon_{r}$ and the surface material conductivity $\sigma_{e}$ are known. Therefore, there is only one unknown in Equation (47), reducing it to a one-dimensional search, and the MUSIC dimension reduction process is completed.

Finally, we use Equation (23) to convert the estimated low-elevation angle of the target after the spectral peak search into the height H of the target.

It should be noted that relative dielectric constant $\varepsilon_{r}$ and surface material conductivity $\sigma_{e}$ cannot be accurately measured in a complex position scene, and there are certain errors. In this situation, a three-dimensional search process is needed, and an alternative search method [26] can be used to alleviate the above problems, but this is not the core content of this paper, so it is not expanded upon here.

## 6. Computational Complexity

In this section, we analyze the computational complexity of the three algorithms and compare them with each other. The computational complexity of the different algorithms is shown in Table 2, where N represents the number of antenna elements, T represents the number of snapshots of the received signals, and n represents the number of DOA angle searches. As we can see from Table 2, the computational complexity of the MPS-GMUSIC algorithm is only $2N^{2}$ more than that of the PS-GMUSIC algorithm, and the computational complexity of the P-SSV-MUSIC algorithm is at least $\;nN^{2}$ less than the other two algorithms.

## 7. Simulation

As described in this section, some simulations we executed in terms of spectrum estimation, computational complexity, RSME, discrimination success probability and the tracking measurement results of the simulated track to evaluate the proposed algorithm. We adopted the meter wave PSA radar system with N=13, d=λ/2, $f_{0}=150\;\mathrm{MHz}$, λ=2 m, γ=85°, η=170°, $h_{a}=10\;m$, and $R_{r}=100\;\mathrm{km}$. Additionally, we assumed that the reflection dielectric is seawater. As shown in Table 1, the surface material conductivity and relative permittivity can be set as $\sigma_{e}=0.5$ and $\varepsilon_{r}=75$, respectively.

### 7.1. Spectrum Estimation of the Proposed Method

Figure 2 demonstrates the spectrum estimation of the three methods. As we can see in Figure 2, the proposed MPS-GMUSIC and P-SSV-MUSIC algorithms can estimate the low-elevation angle of the target, and the overall effect is better than that of the PS-GMUSIC algorithm. However, the elevation estimation results of the MPS-GMUSIC algorithm are only closer to the actual low-elevation angle value; the estimated angle of the P-SSV-MUSIC algorithm is the same as the actual low-elevation angle value, and its spectral peak is sharper, indicating that the method has a better angular resolution, making it more accurate in practical applications.

### 7.2. Computational Complexity of the Proposed Method

In order to compare the computational complexity of the three algorithms more intuitively, we set the number of antenna elements between 2 and 30. In addition, since the search interval set by Simulation 7.1 is $0.01^{\circ }$, the number of DOA angle searches is n=1000. Figure 3 shows the computational complexity of the three algorithms with respect to number of antenna elements. It can be seen that the computational complexity of the P-SV-MUSIC algorithm is significantly lower than that of the other two algorithms.

### 7.3. RMSE of the Proposed Method

In order to assess the angle estimation performance of the proposed algorithm, this section presents the RMSE simulations.

#### 7.3.1. RMSE with Respect to the SNR

We assumed that the SNR between −10 dB and 10 dB. Figure 4 shows the RMSE of the elevation with respect to the SNR when $\theta_{d}$ is $1^{\circ }$ and $2^{\circ }$. As we can see in Figure 4, the estimation accuracy of the MPS-GMUSIC algorithm and P-SSV-MUSIC algorithm is improved when there is an increase in the SNR, which is consistent with the expected results. The accuracy of the MPS-GMUSIC algorithm cannot exceed that of the PS-GMUSIC algorithm when $\theta_{d}$ is $1^{\circ }$. However, the P-SSV-MUSIC algorithm is superior to the other two algorithms in terms of the angle measurement accuracy and height measurement accuracy within the given SNR range, regardless of whether $\theta_{d}$ is $1^{\circ }$ or $2^{\circ }$, and it is an order of magnitude higher. This proves that the P-SSV-MUSIC algorithm has excellent angle measurement accuracy.

Figure 5 shows the RMSE of the elevation with respect to the SNR of the two-polarization parameters and its CRB using the P-SSV-MUSIC algorithm, which is indicates that the algorithm accurately estimates the two-dimensional polarization parameters and can be close to the optimal estimation performance. The CRB corresponding to the direct wave angle and the two-polarization parameters and the deduction process are included in Appendix A.

#### 7.3.2. RMSE with Respect to the Snapshot Number

We assumed that the snapshot number is between 10 and 250 . Figure 6 shows the RMSE of the elevation with respect to the snapshot number when $\theta_{d}$ is $1^{\circ }$ and $2^{\circ }$. As we can see in Figure 6, the estimation accuracy of the MPS-GMUSIC algorithm and P-SSV-MUSIC algorithm improves when there is an increase in the snapshot number, which is consistent with the expected results. The accuracy of the MPS-GMUSIC algorithm cannot exceed that of the PS-GMUSIC algorithm when $\theta_{d}$ is $1^{\circ }$. However, the P-SSV-MUSIC algorithm is superior to the other two algorithms in terms of angle measurement accuracy and height measurement accuracy within the given snapshot number range, regardless of whether $\theta_{d}$ is $1^{\circ }$ or $2^{\circ }$, and it is an order of magnitude higher.

#### 7.3.3. RMSE with Respect to the Phase Aberration

We assumed that the phase aberration is between $0^{\circ }$ and $45^{\circ }$. Figure 7 shows the RMSE of elevation with respect to the phase aberration when $\theta_{d}$ is $1^{\circ }$ and $2^{\circ }$. As we can see in Figure 7, as the phase aberration increases, the angle RSME of the three algorithms increases, and the performance of the algorithm decreases. However, the RSME of the P-SV-MUSIC algorithm is lower than that of the PS-GMUSIC algorithm and the MPS-GMUSIC algorithm, regardless of whether $\theta_{d}$ is $1^{\circ }$ or $2^{\circ }$, which proves that it has higher measurement accuracy.

### 7.4. Discrimination Success Probability of the Proposed Method

In this simulation, we set the search angle to between $0.1^{\circ }$ and $4^{\circ }$. In addition, we consider that the discrimination is successful when the spectrum estimation produces obvious peaks and when the estimated angle is less than or equal to $0.05^{\circ }$. Figure 8 shows the discrimination success probability of the three algorithms with respect to the search angle when the SNR is 15 dB and 20 dB. The simulation results show that the P-SSV-MUSIC algorithm has a higher discrimination success probability and a lower discrimination threshold.

### 7.5. Tracking Measurement Results of Simulated Track

In this simulation, we set the target’s range to be between 50 km and 100 km. The elevation measurement results of the target for the three algorithms are shown in Figure 9. The height measurement results of target for the three algorithms are shown in Figure 10. Compared to the real elevation angle and the real height of the target, we found that the angle and height estimation results of the P-SSV-MUSIC algorithm are almost consistent with the actual value. The angle and height estimation results of the PS-GMUSIC algorithm and MPS-GMUSIC algorithm begin to deviate from the actual values after 85 km, but the MPS-GMUSIC algorithm demonstrates less deviation than the PS-GMUSIC algorithm.

Figure 11 shows the error results of the target elevation measurements for the three algorithms. We can see intuitively in Figure 11 that the error results of the target elevation measurements of the three algorithms are within a small range when the horizontal distance lies before 85 but that large errors begin to occur after 85. It increases as the distance of the target increases, which is consistent with the expected results. Additionally, the error results of the target elevation measurements of the P-SSV-MUSIC algorithm are within the range of $\pm 0.01^{\circ }$ when the horizontal distance is 85–100. The error results of the target elevation measurements of the MPS-GMUSIC algorithm are within the range of $\pm 0.02^{\circ }$ when the horizontal distance is 85–100. The error results of the target elevation measurements of the PS-GMUSIC algorithm are within the range of $\pm 0.12^{\circ }$ when the horizontal distance is 85–100. Figure 12 shows the error results of the target height measurements for the three algorithms. It is clear that the error results of the target height measurements for the P-SSV-MUSIC algorithm are within 25 m and that those of the MPS-GMUSIC algorithm are within 50 m. However, the maximum error of the PS-GMUSIC algorithm is close to 200 m. These indicate that the P-SSV-MUSIC algorithm has better angle and height estimation performance and that it can realize precise height measurements of meter wave radar.

## 8. Conclusions

In this paper, the PS-GMUSIC algorithm is introduced firstly. Then, we propose the MPS-GMUSIC algorithm, which adds forward and backward spatial smoothing processing after polarization smoothing processing comparing with PS-GMUSIC algorithm. However, its improvement of low angle estimation performance is not obvious and cannot estimate the polarization parameters either. Therefore, we propose the P-SSV-MUSIC algorithm finally, which can obtain the elevation and polarization parameters of the target simultaneously, but needs the a priori reflection coefficient. The simulation results show that the proposed algorithms can effectively estimate the target elevation angle in multipath environment. The MPS-GMUSIC algorithm has better performance in height measurement than the polarization smoothed generalized MUSIC algorithm, and the computational complexity of the two algorithms is not much different. In addition, compared with the PS-GMUSIC algorithm and the MPS-GMUSIC algorithm, the P-SSV-MUSIC algorithm can achieve higher height measurement accuracy and discrimination success probability, and has lower computational complexity and resolution threshold. For future work, our focus will be on height measurement of meter wave polarization-sensitive array radar in undulating terrain and the scene of multi-target.

## Figures and Tables

**Figure 1 sensors-22-07298-f001:**
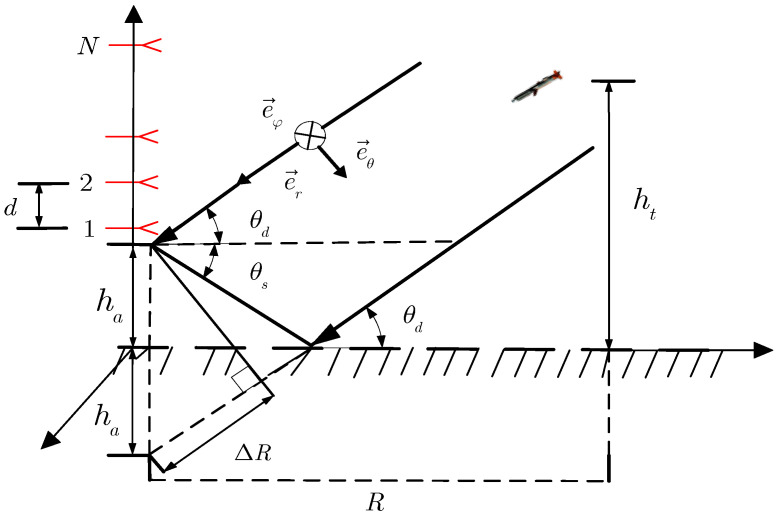
Multipath signal model of meter wave PSA.

**Figure 2 sensors-22-07298-f002:**
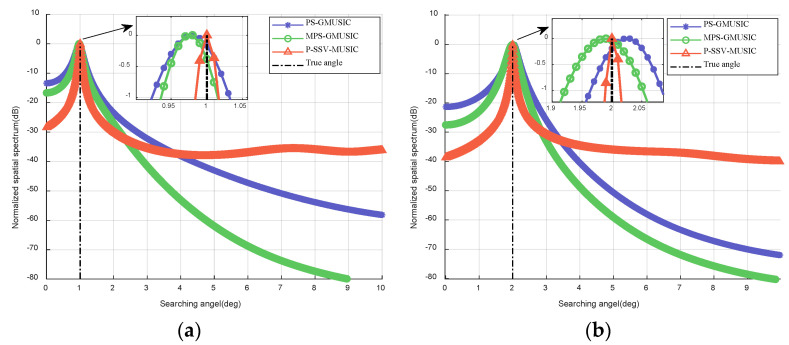
Spectrum estimation of the three methods when SNR=20 dB and T=100: (**a**) $\theta_{d}=1^{\circ }$; (**b**) $\theta_{d}=2^{\circ }$.

**Figure 3 sensors-22-07298-f003:**
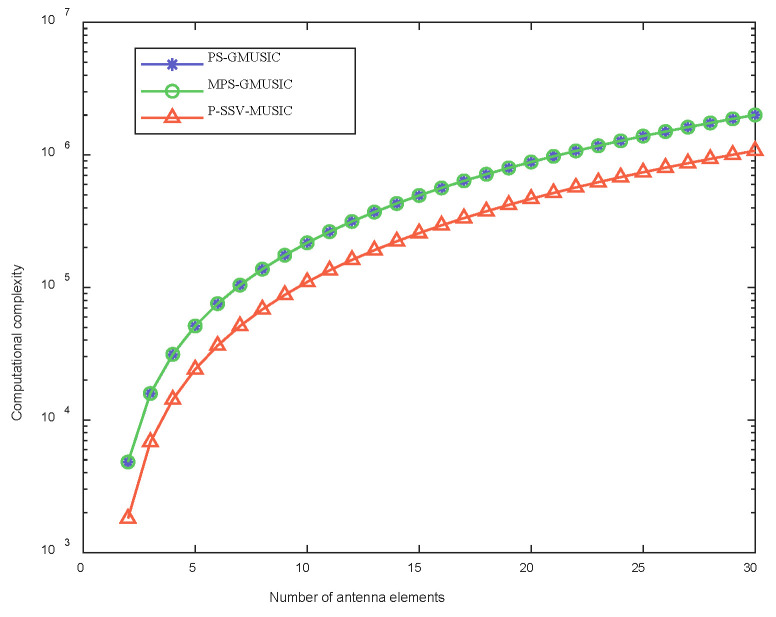
Computational complexity with respect to the number of antenna elements.

**Figure 4 sensors-22-07298-f004:**
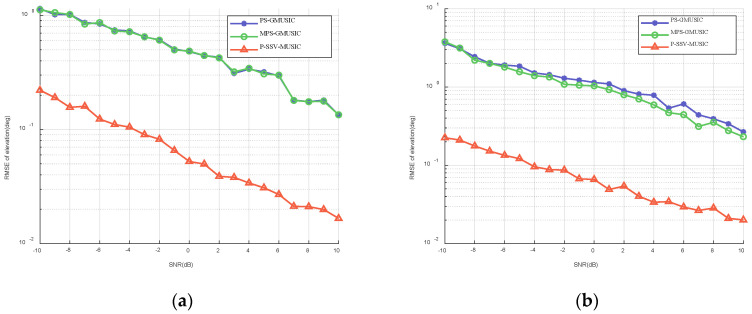
RMSE of elevation with respect to the SNR when T=100: (**a**) $\theta_{d}=1^{\circ }$; (**b**) $\theta_{d}=2^{\circ }$.

**Figure 5 sensors-22-07298-f005:**
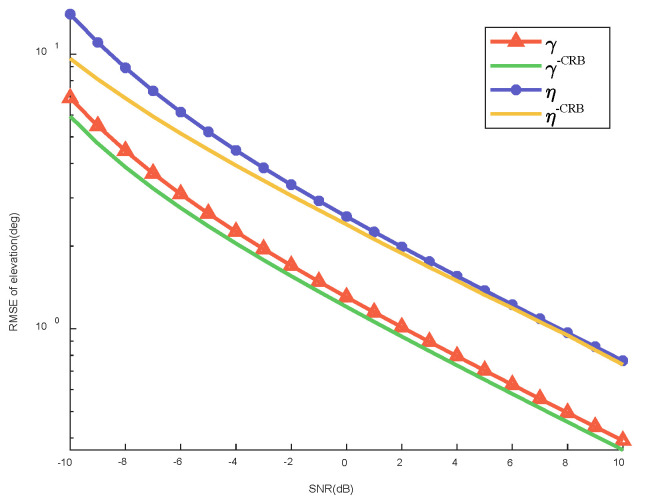
RMSE of elevation with respect to the SNR of the two-polarization parameters and their CRBs using the P-SSV-MUSIC algorithm.

**Figure 6 sensors-22-07298-f006:**
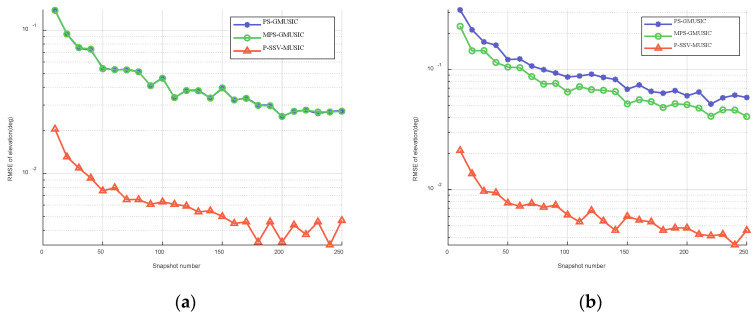
RMSE of elevation with respect to the snapshot number when SNR=20 dB: (**a**) $\theta_{d}=1^{\circ }$; (**b**) $\theta_{d}=2^{\circ }$.

**Figure 7 sensors-22-07298-f007:**
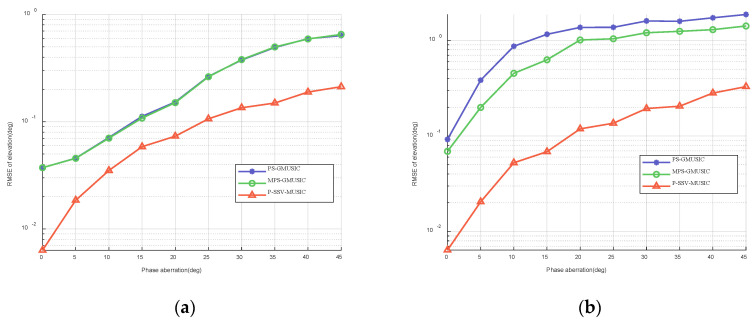
RMSE of elevation with respect to the phase aberration when SNR=20 dB and T=100: (**a**) $\theta_{d}=1^{\circ }$; (**b**) $\theta_{d}=2^{\circ }$.

**Figure 8 sensors-22-07298-f008:**
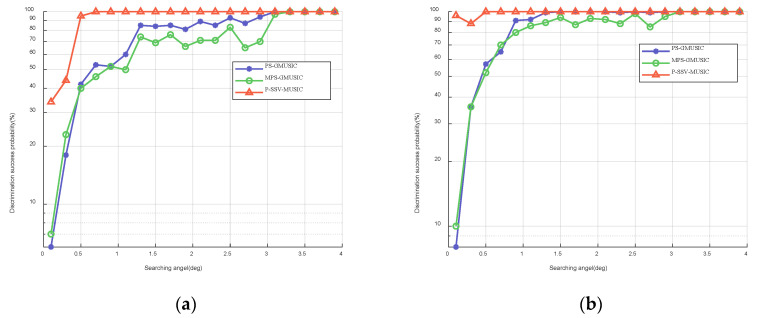
Discrimination success probability with respect to the search angel when T=100: (**a**) SNR=15 dB; (**b**) SNR=20 dB.

**Figure 9 sensors-22-07298-f009:**
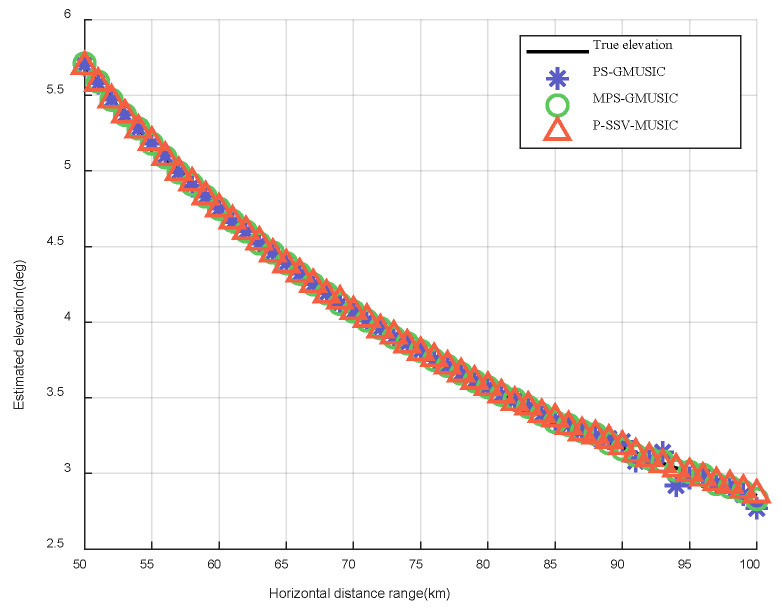
Elevation measurement results of target.

**Figure 10 sensors-22-07298-f010:**
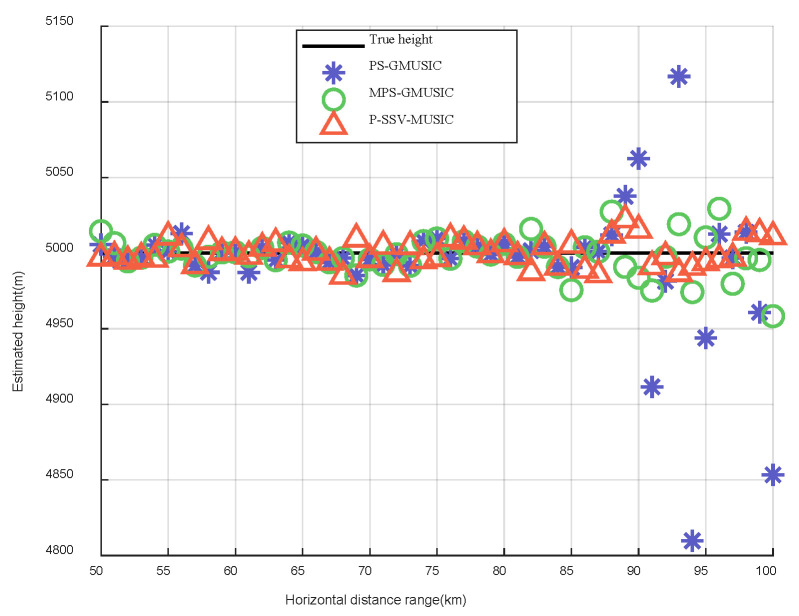
Height measurement results of target.

**Figure 11 sensors-22-07298-f011:**
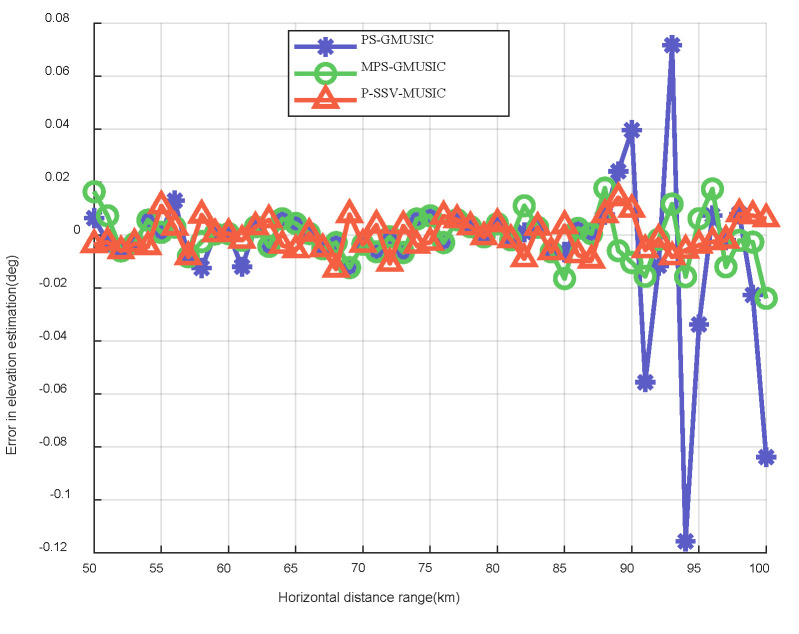
Error results of target elevation measurement.

**Figure 12 sensors-22-07298-f012:**
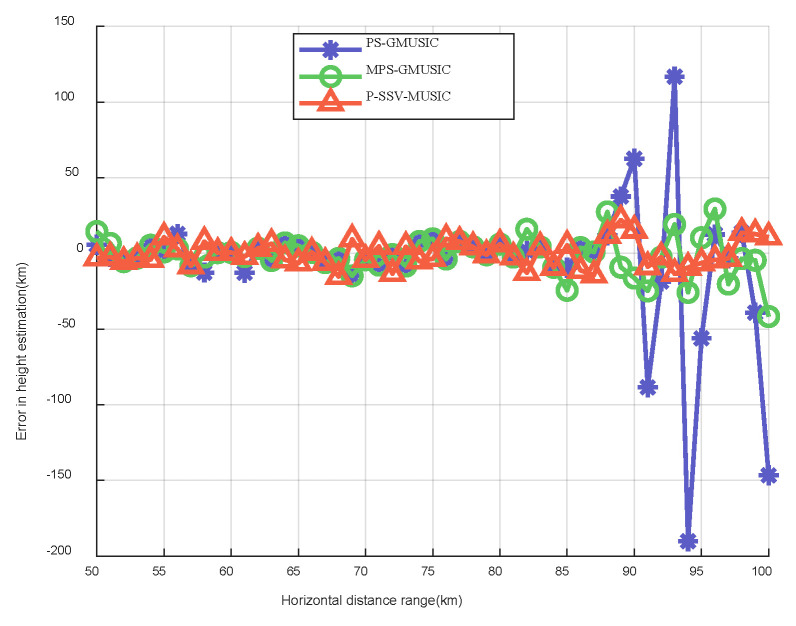
Error results of target height measurement.

**Table 1 sensors-22-07298-t001:** Relative dielectric constant $\varepsilon_{r}$ and surface material conductivity $\sigma_{e}$ under different terrains.

Index	Dielectric	Relative Permittivity	Surface Material Conductivity
1	Good soil (wet soil)	25	0.02
2	General soil	15	0.005
3	Poor soil (dry soil)	3	0.001
4	Snow and ice	3	0.001
5	Freshwater	81	0.7
6	Seawater	75	0.5

**Table 2 sensors-22-07298-t002:** Computational complexity of three algorithms.

Algorithm Name	Computational Complexity
PS-GMUSIC	$O\left(N^{3}+2N^{2}T+n\left(2N^{2}-4\right)\right)\;$
MPS-GMUSIC	$O\left(N^{3}+2N^{2}T+2N^{2}+n\left(2N^{2}-4\right)\right)\;$
P-SSV-MUSIC	$O\left(N^{3}+2N^{2}T+n\left(N^{2}-N-1\right)\right)\;$

## Data Availability

Not applicable.

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
