# Peer review of "Meter Wave Polarization-Sensitive Array Radar for Height Measurement Based on MUSIC Algorithm"

_sensors, 2022, doi:10.3390/s22197298_

Round 1
Reviewer 1 Report
The paper proposes the modified polarization smoothing generalized MUSIC algorithm and the polarization synthesis steering vector MUSIC algorithm.Simulation shows that the P-SSV-MUSIC algorithm is superior to PS-GMUSIC algorithm and MPS-GMUSIC algorithm in accuracy and computational complexity, and the accuracy can be increased by an order of magnitude. This paper shows some new research results, but there are still some problems about the writing. I have some questions as follows:
1. The proposed modified polarization smoothing generalized MUSIC algorithm is not novel enough, which is the combination of previous algorithms.
2. The computational complexity of three algorithms has no corresponding simulation.
3. The content of “the three algorithm” in 7.3 should be “the three algorithms”.
4. In the fifth reference, the period before 1985 should be changed to comma.
5. In (5), there exists some errors. Please revised it.
Author Response
Point 1: The proposed modified polarization smoothing generalized MUSIC algorithm is not novel enough, which is the combination of previous algorithms.
Response 1: The proposed modified polarization smoothing generalized MUSIC algorithm is indeed a combination of previous algorithms. Therefore, it is used as a comparison algorithmit. The focus of this paper is the proposed P-SSV-MUSIC algorithm.
Point 2: The computational complexity of three algorithms has no corresponding simulation.
Response 2: It has been added in the manuscript.
Point 3: The content of “the three algorithm” in 7.3 should be “the three algorithms”.
Response 3: It has been modified in the manuscript.
Point 4: In the fifth reference, the period before 1985 should be changed to comma.
Response 4: It has been modified in the manuscript.
Point 5: In (5), there exists some errors. Please revised it.
Response 5: It has been modified in the manuscript.
In view of the reviewers' evaluation of English language and style, we used the language editing service of MDPI.
Reviewer 2 Report
The authors present the Meter Wave Polarization Sensitive Array Radar for Height Measurement based on MUSIC Algorithm.
1. There are several algorithms proposed in the state of the art literature, why the authors only focused MUSIC Algorithm?
2. Did the authors implemented the proposed concept in real time environment? it is suggested to implement the proposed work in real time environment.
3. The novelty is not well explained. it is recommended to highlight the innovative points of the work in the bullets.
4. The included references are not up-to-date. The latest reference should be added.
5. it would be better if the authors include the latest comparison analysis in the manuscript.
Author Response
The word file provides a point-by-point response to the reviewer’s comments.

Reviewer 3 Report
1. The contribution of the paper should be highlighted.
2. Applying MUSIC for estimation is somewhat common now, given its well-developed theories. I wonder what is new here? Also, the impact of the number of targets, known or unknown, on MUSIC is worth discussing and being shown in simulations.
3. The authors assume that the array is ULA. Can the proposed method be applied for other geometrical array such as UPA, even arbitrary geometrical array?
4. How can you measure the covariance matrix R^{ps} in practice? And how do you measure it in the simulation? please explain it.
5. Do you have some error in measuring the covariance matrix? What is the effect of this error on your algorithm?
6. How much is the resolution of your proposed method? Please discuses about it either by statements or the simulations.
7. There are some writting errors in this manuscript,such as Eq. 5. Please carefully correct them before you submit the revised paper.
8. A number of references closely related with the proposed method should be reviewed in the introduction of the manuscript, e.g.,
[R1] "Multi-mode OAM Radio Waves: Generation, Angle of Arrival Estimation and Reception With UCAs".
[R2] "Joint OAM Radar-Communication Systems: Target Recognition and Beam Optimization".
[R3] "AoA Estimation for OAM Communication Systems With Mode-Frequency Multi-Time ESPRIT Method".
Author Response

(The authors gave the same response as above.)

Reviewer 4 Report
Comments to the authors:
1. The paper must be comprehensively revised for coherency and checked for English writing. The paper is not easy to follow and understand.
2. Line 91, in section 1, please clearly specify the problems that are addressed by the proposed method.
3. Page 4, line 129, how did you derive alpha? Is there any assumption?
4. Equation (5) is not clear.
5. Do you have any assumption in (7)? Is it at a specific frequency?
6. Please define the matrix B in page 6, line 176.
7. The performance must be assessed in presence of rough surfaces and phase aberrations caused by these surfaces in the received signal.
8. What is the performance if more than one target is present in the scene?
9. Please assess the performance of P-SSV-MUSIC if there a level of error in estimation of the reflection coefficient.
Author Response

(The authors gave the same response as above.)

Round 2
Reviewer 2 Report
In the last round of reviews, i suggested you to explain the novelty and implement your work in the real time environment. did you incorporate the suggested changes? I only can read your response letter and its not sufficient to convince the readers and for the high quality articles.
It shows the seriousness of the authors to their scientific work! Further, i am suggesting you to incorporate the comments and submit it to another suitable journal related to your area of interest.
Author Response
Dear Reviewer:
Thank you very much for your suggestions. All your suggestions are very important, and they are of great guiding significance for our paper writing and scientific research. We regret that the last reply did not satisfy you, so this time we answered your questions in detail, made relevant modifications in the manuscript, and resubmitted the manuscript to meet your requirements for high-quality articles and show our seriousness in scientific work.

Reviewer 3 Report
My comments are well replied and the corresponding contents are improved satisfactorily.
Author Response
Thank you for your suggestions. All your suggestions are very important. They have important guiding significance for my thesis writing and scientific research work. Also thank you for your recognition, I will continue to work hard. Finally, blessings to you! May all the best things be around you forever!
Reviewer 4 Report
The paper could be accepted in the current form. However, the response regarding the performance of the method in presence of error in estimation of the reflection coefficient is not satisfactory.
Author Response
Thanks very much for taking your time to review this manuscript. We really appreciate all your generous comments and suggestions! All your suggestions are very important. They have important guiding significance for my thesis writing and scientific research work. It's very regretful that the response regarding the performance of the method in presence of error in estimation of the reflection coefficient is not satisfactory. We are extremely grateful to you for pointing out this problem. We will deepen it in the follow-up work in combination with this opinion. Finally, blessings to you! May all the best things be around you forever!
Round 3
Reviewer 2 Report
Thanks for incorporating the suggestions. i have still following concerns, pls incorporate it in the revised manuscript.
1. pls write the novelty points and key contributions of the work right after the line 109 of the article.
2. pls add future directions of the study in the conclusion section.
Author Response
Dear Reviewer:
Thank you very much for your suggestions again. In response to your suggestions, I revised the manuscript at the corresponding location. Finally, blessings to you! May all the best things be around you forever!
Kind regards,
Mr.Wang